# Detection of Cassava Component in Sweet Potato Noodles by Real-Time Loop-mediated Isothermal Amplification (Real-time LAMP) Method

**DOI:** 10.3390/molecules24112043

**Published:** 2019-05-29

**Authors:** Deguo Wang, Yongzhen Wang, Kai Zhu, Lijia Shi, Meng Zhang, Jianghan Yu, Yanhong Liu

**Affiliations:** 1Key Laboratory of Biomarker Based Rapid-Detection Technology for Food Safety of Henan Province, Xuchang University, Xuchang 461000, China; wangdg666@126.com (D.W.); wangdg666@aliyun.com (Y.W.); 15649832270@163.com (K.Z.); shilijia19940908@163.com (L.S.); peterrabbit@yeah.net (M.Z.); yjh18838466861@163.com (J.Y.); 2Molecular Characterization of Foodborne Pathogens Research Unit, Eastern Regional Research Center, Agricultural Research Service, United States Department of Agriculture, Wyndmoor, PA 19038, USA

**Keywords:** loop-mediated isothermal amplification (LAMP), adulterated sweet potato noodles, cassava-derived ingredients, internal transcribed spacer (ITS)

## Abstract

Sweet potato (*Ipomoea batatas*) noodles are a traditional Chinese food with a high nutritional value; however, starch adulteration is a big concern. The objective of this study was to develop a reliable method for the rapid detection of cassava (*Manihot esculenta*) components in sweet potato noodles to protect consumers from commercial adulteration. Five specific Loop-mediated Isothermal Amplification (LAMP) primers targeting the internal transcribed spacer (ITS) of cassava were designed, genomic DNA was extracted, the LAMP reaction system was optimized, and the specificity of the primers was verified with genomic DNA of cassava, *Ipomoea batatas*, *Zea mays*, and *Solanum tuberosum*; the detection limit was determined with a serial dilution of adulterated sweet potato starch with cassava starch, and the real-time LAMP method for the detection of the cassava-derived ingredient in sweet potato noodles was established. The results showed that the real-time LAMP method can accurately and specifically detect the cassava component in sweet potato noodles with a detection limit of 1%. Furthermore, the LAMP assay was validated using commercial sweet potato noodle samples, and results showed that 57.7% of sweet potato noodle products (30/52) from retail markets were adulterated with cassava starch in China. This study provides a promising solution for facilitating the surveillance of the commercial adulteration of sweet potato noodles from retail markets.

## 1. Introduction

Sweet potato (*Ipomoea batatas*) is a dicotyledonous plant that belongs to the family of *Convolvulaceae*. The roots of sweet potato are starchy, sweet-tasting, and tuberous vegetables. Sweet potato noodles are favored by many consumers due to this food’s ease of cooking, good nutritional value, and easy digestion. There is a huge market for this food product in Asia, including China, Japan, and Korea. The food labels of noodle products are required to specify the starch source specifically to prevent adulteration and to protect consumers. However, the mixture of low-priced starch into high-priced starch is still very common. For example, cassava starch is often incorporated into sweet potato starch during the production of sweet potato noodles; therefore, a reliable method for the identification of starch species is critical for the surveillance of commercial adulteration [1].

At present, there have been some reports on the detection of starch adulteration, including methods based on physical properties (cohesiveness, extension, cutting behavior, cooking loss, and swelling index) and sensory evaluation [2], which are vulnerable to subjective factors. Methods based on near-infrared spectroscopy have a low sensitivity [3,4,5,6,7], and DNA-based methods such as Polymerase Chain Reaction (PCR) and DNA barcoding are very effective but rely on the requirements of professional skill and expensive instrumentation [8,9,10,11]. There is a need to develop a rapid, sensitive, and cost-effective method for the identification of adulterated starch, especially for sweet potato noodles processed at a high temperature.

Loop-mediated isothermal amplification (LAMP) has been developed to amplify nucleic acids under isothermal conditions, and it is a very specific, sensitive, and rapid technology [12]. The method has been reported to identify GMO maize starch [13]. The LAMP reaction is easily performed, and it is highly specific to the target sequence because six independent primes recognize the target sequence in the initial stage, and four independent primers are used to amplify the target sequence in the later stage of the LAMP reaction [14]. The amplification efficiency of the LAMP method is extremely high because it is an isothermal reaction with no ramp time for thermal change [15]. The objective of this study was to develop a reliable real-time LAMP method for the rapid detection of cassava-derived ingredients, to protect consumers from commercial adulteration. The most common material for sweet potato noodle fraud is starch of cassava (*Manihot esculenta*). In addition, the starch from *Zea mays* or *Solanum tuberosum* is sometimes used as the counterfeit material. The specificity of a newly developed real-time LAMP method for the rapid detection of cassava-derived ingredients was tested with starch from cassava (*Manihot esculenta*), *Ipomoea batatas*, *Zea mays* and *Solanum tuberosum*.

## 2. Results

### 2.1. Primer Specificity for the LAMP Assay

As shown in Figure 1, the length of the target sequences from *Manihot esculenta* was 193 bp. The identity of the sequences with the corresponding sequences of *Ipomoea batatas* (GenBank: MH792118.1), *Zea mays* (GenBank: KU182536.1), and *Solanum tuberosum* (GenBank: AY875827.1) was 79.91%, blasted with DNAMAN (Lynnon Biosoft, San Ramon, USA); thus, the real-time LAMP assay using the cassava primers had a high specificity.

### 2.2. Optimization of the Real-Time LAMP Reaction Temperature

The LAMP reaction with the newly designed primers was carried out at 56 °C and 58 °C for 60 min, and as shown in Figure 2, it resulted in the amplification of all positive controls (DNA from *Manihot esculenta*) along with negative results for all negative control (water only) reactions. Given the amplification efficiency and specificity, 58 °C was chosen as the most suitable reaction temperature.

### 2.3. Specificity of the Real-Time LAMP Assay

The specificity of the LAMP assay was tested with starch samples of *Manihot esculenta*, *Ipomoea batatas*, *Zea mays,* and *Solanum tuberosum*. The reason for this type of sample selection was that, alongside the most common material for sweet potato noodle fraud, starch from *Zea mays* or *Solanum tuberosum* is sometimes used as the counterfeit material. As shown in Figure 3, both cassava starch samples were successfully detected, while the starch samples from the other species were negative.

### 2.4. Sensitivity of the Real-Time LAMP Assay

The sensitivity of the LAMP assay was determined with genomic DNA extracted from sweet potato starch mixed with different proportions of cassava starch at 58 °C for 60 min. The detection limit was found to be the sweet potato starch mixed with 1% cassava starch, as shown in Figure 4.

### 2.5. Validation of the LAMP Assay for Detection of Cassava Starch Contamination of Sweet Potato Noodle Samples in China

Out of 52 sweet potato noodle samples collected from Chinese retail markets that were tested, 30 were found to be positive for cassava starch using the newly established real-time LAMP assay, indicating that fraud of sweet potato noodles with cassava starch may be very common in China.

## 3. Discussion

Since the LAMP technique was first reported in 2000 [12], researchers have explored its usage for analysis in various fields due to its excellent properties, including rapidity, sensitivity and specificity [16,17]; furthermore, the reports have been cited more than 5568 times, as listed by Google Scholar up to May 2019. The most notable advantage of LAMP is its rapidity, since LAMP can be performed in just 30 min because of the absence of a ramp time for thermal change [17,18]. The sensitivity of LAMP was not affected by the presence of non-target DNA in the samples, and the method was also more tolerant to well-known PCR inhibitors such as blood, serum and food ingredients [19]. LAMP has been developed to target various meat species, such as porcine, chicken, horse, and ostrich and had a sensitivity as low as 0.1 pg/μL [20]. The amplification specificity of LAMP was extremely high because the primers bind six independent sequences of the target DNA [15,19]. These properties made it attractive for us to explore the use of LAMP for the identification of sweet potato noodle adulteration.

Since starch contains a high carbohydrate content, and sweet potato starch was subjected to a heat treatment, it was difficult to extract enough high-quality genomic DNA from the noodles for the use of sweet potato noodle fraud testing with the DNA-based method. The improved CTAB method was used as the DNA extraction protocol in this study, and the internal transcribed spacer (ITS), with high copy number, was selected as the target sequence, significantly increasing the sensitivity and practicability of the newly established real-time LAMP assay.

There is no national standard or industry standard for sweet potato noodles in China, and the current national standard (GB/T 23587-2009 Vermicelli) does not have a method to distinguish noodles made of sweet potato starch from those made with low-cost cassava starch; consequently, sweet potato noodle fraud with cassava starch is very common in China. The adulteration of sweet potato noodles economically impacts sweet potato plantations and seriously impacts consumers. The advantages of the real-time LAMP assay are its cost-effectiveness, simplicity, high specificity and sensitivity, which make it suitable for the surveillance of commercial sweet potato noodle products that are adulterated with cassava starch.

## 4. Materials and Methods

### 4.1. Primer Design for LAMP Assay

A set of LAMP primers targeting the internal transcribed spacer (ITS) of *Manihot esculenta* (GenBank accession No. MK114629.1) were designed using PrimerExplorer V5 (http://primerexplorer.jp/e/) and Oligo 7 (Molecular Biology Insights, Inc. Colorado Springs, CO, USA). The primer sequences are listed in Table 1.

### 4.2. Genomic DNA Isolation

Genomic DNA from the starch samples of *Manihot esculenta*, *Ipomoea batatas*, *Zea mays*, and *Solanum tuberosum* was extracted according to the method of Rajoo et al. [9]. Briefly, starch powders (2 g), homogenized with 10 mL of 3% cetyltrimethylammounium bromide (CTAB) buffer containing 0.3% β-mercaptoethanol, were transferred to 50 mL tubes. After an incubation at 60 °C for 2 h with intermittent shaking, 3 mL of phenol: chloroform: isoamylalcohol (25:24:1) were added, mixed by inversion, and centrifuged at 8000× *g* for 15 min at 4 °C. The aqueous phase was mixed with 7 mL of chloroform: isoamylalcohol (24:1) by inversion for 15 min and centrifuged at 8000× *g* for 15 min at 4 °C. The aqueous phase was transferred to a fresh tube, and 0.6 volume of ice-cold isopropanol was added and incubated at −20 °C for 2 h to precipitate the DNA. After centrifugation at 4000× *g* for 5 min, the pellets were washed with 70% ethanol, dried in the air and dissolved in sterile double distilled water. RNase was added to the DNA solution at a concentration of 10 mg/mL, followed by incubation at 37 °C for 1 h. An equal volume of phenol: chloroform: isoamylalchohol (25:24:1) was added to the DNA solution, mixed for 15 min, and centrifuged at 10,000× *g* for 15 min at 4 °C. The aqueous phase was re-extracted for phenol. An equal volume of 100% ice-cold ethanol was added to the aqueous phase and incubated at −20 °C for 2 h to precipitate the DNA. After centrifugation at 4000× *g* for 5 min, the pellets were washed with 70% ethanol to remove salts. The pellets were air dried and dissolved in 500 µL double distilled water. Two hundred microliters of 30% PEG 8000 were added to 500 µL of the DNA sample, mixed, incubated at room temperature for 30 min, and centrifuged at 14,000× *g* for 15 min. The pellets were washed with 80% ethanol, air-dried, and dissolved in nuclease free water or Tris-EDTA (10:1) buffer. The samples were frozen at −20 °C until further use. The noodle samples were homogenized using mortars and pestles before the DNA extraction.

### 4.3. Optimization of Real-Time LAMP Reaction Temperature

The real-time LAMP assay was performed in a 10-μL reaction mixture containing 0.8 mM each of forward inner primer (FIP) and backward inner primer (BIP), 0.2 mM each of forward outer primer (F3) and backward outer primer (B3), 0.4 mM of forward loop primer (LF), 1.0 mM dNTPs, 20 mM Tris-HCl (pH 8.8), 10 mM KCl, 10 mM (NH_4_)_2_SO_4_, 6 mM MgSO_4_, 0.1% Triton X-100, 7.5% DMSO [21], 1 × EvaGreen, 1 × Rox, 10 ng DNA template of *Manihot esculenta*, and 3.2 U Bst 2.0 WarmStart DNA polymerase (New England Biolabs, Beverly, Mass., USA.) [22]. The reaction mixture was heated at 56 °C or 58 °C for 60 min (30 s per cycle), and a melt curve was obtained using a StepOne^TM^ System (Applied Biosystems, Foster City, CA, USA).

### 4.4. Specificity Determination of the Real-Time LAMP Assay

Genomic DNA of *Manihot esculenta*, *Ipomoea batatas*, *Zea mays*, and *Solanum tuberosum* was used for determining the specificity of our developed LAMP assay, and the amount of genomic DNA template used was 1 ng per reaction.

### 4.5. Sensitivity Determination of the Real-Time LAMP Assay

The sweet potato starch was mixed proportionally with the cassava starch; the mass percent of cassava starch was 10%, 5%, 1%, and 0.1%, and genomic DNA was extracted according to the method of Rajoo et al. [9]. The detection limit was determined using the above real-time LAMP assay.

### 4.6. Analysis of Sweet Potato Noodle Samples from Retail Markets

A total of 52 samples labeled sweet potato noodles were collected from Chinese markets; the samples were homogenized, their DNA was extracted according to the method of Rajoo et al. [9], and the adulteration with the cassava component was tested with the real-time LAMP assay.

## Figures and Tables

**Figure 1 molecules-24-02043-f001:**
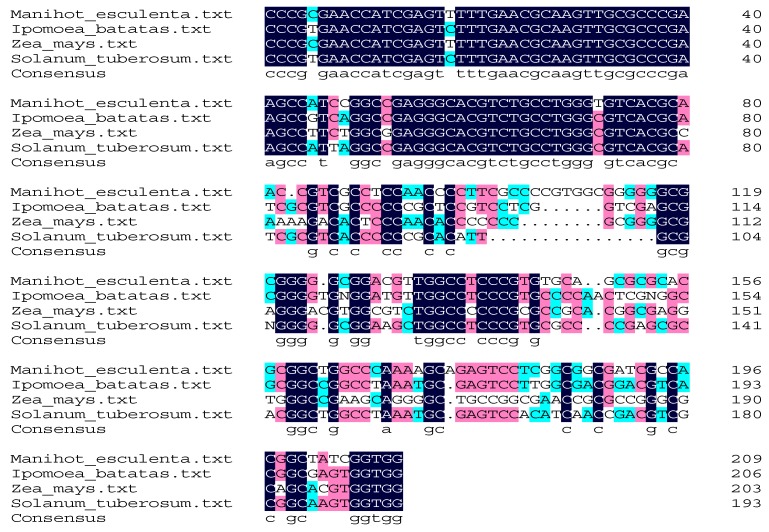
The alignment of the target sequences from *Manihot esculenta* with the corresponding sequences of *Ipomoea batatas*, *Zea mays*, and *Solanum tuberosum*.

**Figure 2 molecules-24-02043-f002:**
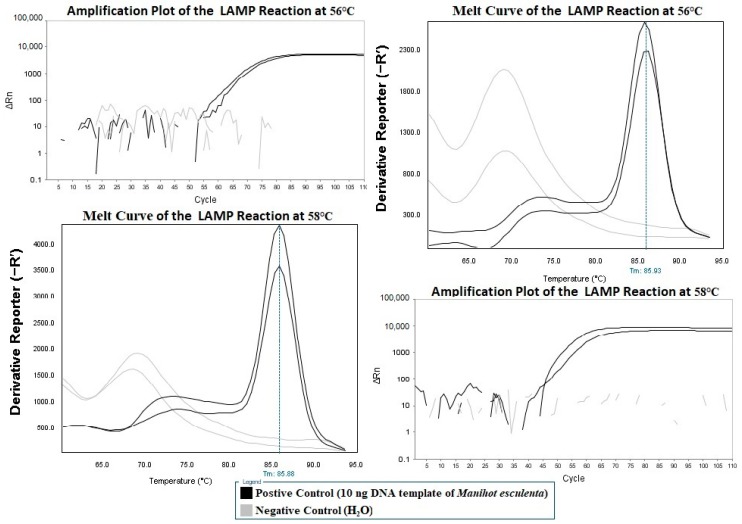
The LAMP reactions at 56 °C and 58 °C.

**Figure 3 molecules-24-02043-f003:**
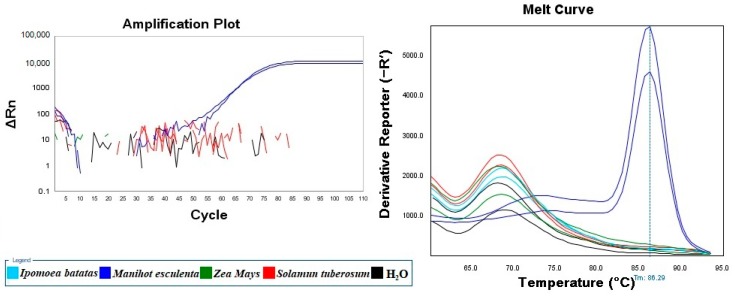
Specificity of the Real-time LAMP Assay.

**Figure 4 molecules-24-02043-f004:**
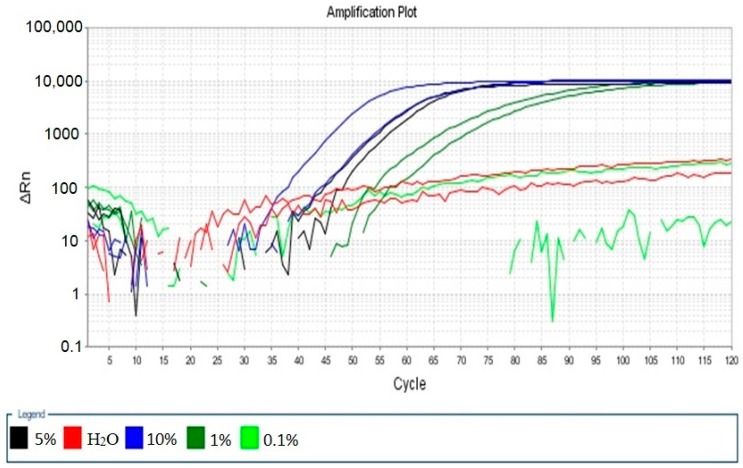
The detection limit of the real-time LAMP Assay. 10%: DNA extracted from sweet potato starch containing 10% cassava starch; 5%: DNA extracted from sweet potato starch containing 5% cassava starch; 1%: DNA extracted from sweet potato starch containing 1% cassava starch; and 0.1%: DNA extracted from sweet potato starch containing 0.1% cassava starch.

**Table 1 molecules-24-02043-t001:** LAMP primers from the Internal Transcribed Spacer (ITS) of *Manihot esculenta*.

Primer	Sequence (5′-3′)
FIP	GGTTGCGTGACACCCAGGCATTTTACGCAAGTTGCGCCCG
BIP	GGACGTTGGCCTCCCGTGTTTTTCGCCGAGGACTCTGCTT
F3	CCCGCGAACCATCGAGTT
B3	ACCACCGATAGCCGTGG
LF	TCGGCCGGATGGCTT

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
