# Peer review of "Detection of Cassava Component in Sweet Potato Noodles by Real-Time Loop-mediated Isothermal Amplification (Real-time LAMP) Method"

_molecules, 2019, doi:10.3390/molecules24112043_

Round 1

Reviewer 1 Report

In this paper, the authors report upon a loop-mediated isothermal amplification (LAMP) DNA high resolution melt assay they have developed to detect cassava (Manihot esculenta) starch in adulterated sweet potato noodles. This is an interesting, concise and generally well-written paper.

Specific comments: Include the genus and species of sweet potato. Write the genomic DNA isolation section (4.2) in past tense. The last two sentences of the discussion are not clear. “Not only this fraud blows the sweet potato plantation heavily but also reduces the interests of consumers seriously. The developed real-time LAMP assay had the advantages of cost-effective, simplicity, high specificity and sensitivity, which will be suitable for surveillance of commercial sweet potato noodle products adulterated with cassava starch.” Suggested edit: Adulterated sweet potato noodles economically impact sweet potato plantations and seriously impact consumers. The advantages of the real-time LAMP assay are its cost-effectiveness, simplicity, high specificity and sensitivity, which makes it suitable for the surveillance of commercial sweet potato noodle products adulterated with cassava starch.

Author Response

Review Report Form 1

Open Review

English language and style

( ) Extensive editing of English language and style required 
( ) Moderate English changes required 
(x) English language and style are fine/minor spell check required 
( ) I don't feel qualified to judge about the English language and style 

Yes

Can be improved

Must be improved

Not applicable

Does the introduction provide sufficient background and include   all relevant references?

(x)

( )

( )

( )

Is the research design appropriate?

(x)

( )

( )

( )

Are the methods adequately described?

(x)

( )

( )

( )

Are the results clearly presented?

(x)

( )

( )

( )

Are the conclusions supported by the results?

(x)

( )

( )

( )

Comments and Suggestions for Authors

In this paper, the authors report upon a loop-mediated isothermal amplification (LAMP) DNA high resolution melt assay they have developed to detect cassava (Manihot esculenta) starch in adulterated sweet potato noodles. This is an interesting, concise and generally well-written paper.

Specific comments:

Include the genus and species of sweet potato.

Answer: Ipomoea batatas.  This has been added in the text.  In addition, the following sentences have been added into the introduction: Sweet potato (Ipomoea batatas) is a dicotyledonous plant that belongs to the family of Convolvulaceae. The roots of sweet potato are starchy, sweet-tasting, and tuberous vegetables.

Write the genomic DNA isolation section (4.2) in past tense.

Answer: The whole paragraph was rewritten and incorporated into the text as the following:

Genomic DNA from the starch samples of Manihot esculenta, Ipomoea batatas, Zea mays, and Solanum tuberosum was extracted according to the method of Rajoo et al [9].  Briefly, starch powders (2 g) homogenized with 10 ml of 3% cetyltrimethylammounium bromide (CTAB) buffer containing 0.3% β-mercaptoethanol, were transferred to 50 ml tubes.  After incubation at 60ºC for 2h with intermittent shaking, 3 ml of phenol: chloroform: isoamylalcohol (25:24:1) were added, mixed by inversion, and centrifuged at 8,000 g for 15 min at 4°C.  The aqueous phase was mixed with 7 ml of chloroform: isoamylalcohol (24:1) by inversion for 15 min,and centrifuged at 8,000 g for 15 min at 4°C.The aqueous phase was transferred to a fresh tube and0.6 volume of ice-cold isopropanol was added and incubated at -20°C for 2 h to precipitate the DNA. After centrifugation at 4,000 g for 5 min, the pellets were washed with 70% ethanol, dried in the air and dissolved in sterile double distilled water. RNase was added to the DNA solution at a concentration of 10 mg/ml, followed by incubation at 37°C for 1 h.An equal volume of phenol: chloroform: isoamylalchohol (25:24 :1) was added to the DNA solution, mixed for 15 min, and  centrifuged at 10,000 g for 15 min at 4°C. The aqueous phase was re-extracted for phenol. An equal volume of 100% ice-cold ethanol was added to aqueous phase and incubated at -20°C for 2 h to precipitate the DNA. After centrifugation at 4000 g for 5 min, the pellets were washed with 70% ethanol to remove salts. The pellets were air dried,and dissolved in 500 ml double distilled water. Two hundred millilitersml of 30% PEG 8000 were added to 500ml of the DNA sample, mixed, incubated at room temperature for 30 min, and centrifuged at 14,000 g for 15 min.  The pellets were washed with 80% ethanol, air-dried, and dissolved in nuclease free water or Tris-EDTA (10:1) buffer, The samples were frozen at -20°C until further use. The noodle samples were ground using motor and pestles before DNA extraction.

The last two sentences of the discussion are not clear. “Not only this fraud blows the sweet potato plantation heavily but also reduces the interests of consumers seriously. The developed real-time LAMP assay had the advantages of cost-effective, simplicity, high specificity and sensitivity, which will be suitable for surveillance of commercial sweet potato noodle products adulterated with cassava starch.” Suggested edit: Adulterated sweet potato noodles economically impact sweet potato plantations and seriously impact consumers. The advantages of the real-time LAMP assay are its cost-effectiveness, simplicity, high specificity and sensitivity, which makes it suitable for the surveillance of commercial sweet potato noodle products adulterated with cassava starch.

Answer: Changes made as suggested in the text.  The following has been added to the discussion: Adulterated of sweet potato noodles economically impact sweet potato plantations and seriously impact consumers. The advantages of the real-time LAMP assay are cost-effectiveness, simplicity, high specificity and sensitivity, which makes it suitable for the surveillance of commercial sweet potato noodle products adulterated with cassava starch.

Submission Date

04 May 2019

Date of this review

13 May 2019 14:08:12

Reviewer 2 Report

This article entitled “Detection of Cassava Component in Sweet Potato Noodles by Real-time Loop-mediated isothermal amplification (Real-time LAMP) Method” was to develop a reliable method for rapid detection of cassava (Manihot esculenta) components in sweet potato noodles to protect consumers from commercial adulteration.  The authors found that the real-time LAMP method developed in this study can accurately and specifically detect the cassava component in sweet potato noodles with a detection limit of 1%.  The work is interesting and might be applied to facilitate the surveillance of the commercial starch adulteration.

Some comments are given as follows.

1.    Line 107: For the abbreviation "CTAB", the full name should be explained the first time it appears in the main text.

2.    The names such as “Manihot esculenta” ,“cassava” or “Manihot esculenta (cassava)” should be mentioned and presented in a more consistent pattern throughout the text.

3.    It was suggested to keep the presentation of unit in a more consistent way throughout the text.  Authors should carefully check it.

4.    The current discussion section is more like an extended description of the results.  It would be better to include more references in the discussion section.  Accordingly, the results should be better discussed on the basis of the existing literature, hence the novelty, feasibility, sensitivity, and advantages of the new method could be discussed more in-depth to fit the objective and topic of this manuscript.

5.    The format of the reference section was inconsistent.  Authors should carefully check it.

Author Response

Review Report Form 2

Open Review

English language and style

( ) Extensive editing of English language and style required 
( ) Moderate English changes required 
( ) English language and style are fine/minor spell check required 
(x) I don't feel qualified to judge about the English language and style 

Yes

Can be improved

Must be improved

Not applicable

Does the introduction provide sufficient background and include   all relevant references?

(x)

( )

( )

( )

Is the research design appropriate?

(x)

( )

( )

( )

Are the methods adequately described?

( )

(x)

( )

( )

Are the results clearly presented?

( )

(x)

( )

( )

Are the conclusions supported by the results?

(x)

( )

( )

( )

Comments and Suggestions for Authors

This article entitled “Detection of Cassava Component in Sweet Potato Noodles by Real-time Loop-mediated isothermal amplification (Real-time LAMP) Method” was to develop a reliable method for rapid detection of cassava (Manihot esculenta) components in sweet potato noodles to protect consumers from commercial adulteration.  The authors found that the real-time LAMP method developed in this study can accurately and specifically detect the cassava component in sweet potato noodles with a detection limit of 1%.  The work is interesting and might be applied to facilitate the surveillance of the commercial starch adulteration.

Some comments are given as follows.

Line 107: For the abbreviation "CTAB", the full name should be explained the first time it appears in the main text.

Answer: changes made as suggested in the text. Cetyltrimethylammounium bromide (CTAB) was added in the text.

The names such as “Manihot esculenta”, “cassava” or “Manihot esculenta (cassava)” should be mentioned and presented in a more consistent pattern throughout the text.

Answer: This issue has been corrected in the text.  Cassava was used for most of the times.

It was suggested to keep the presentation of unit in a more consistent way throughout the text.  Authors should carefully check it.

Answer: This issue has been checked.

The current discussion section is more like an extended description of the results.  It would be better to include more references in the discussion section.  Accordingly, the results should be better discussed on the basis of the existing literature, hence the novelty, feasibility, sensitivity, and advantages of the new method could be discussed more in-depth to fit the objective and topic of this manuscript.

Answer: The following paragraph was added to the discussion with references.

Since the LAMP technique was firstly reported in 2000 [12], researchers have explored its usage for analysis in various fields due to its excellent properties, including rapidity, sensitivity and specificity [16,17], and the reports have been cited more than 5568 times listed by Google Scholar up to May 2019. The most notable advantage of LAMP is its rapidity since LAMP can be performed in just 30 min because of the absence of a ramp time for thermal change [17,18]. The sensitivity of LAMP was not affected by the presence of non-target DNA in samples, and the method was also more tolerant to well-known PCR inhibitors such as blood, serum and food ingredients [19]. LAMP has been developed to target various meat species such as porcine, chicken, horse, and ostrich and had a sensitivity as low as 0.1 pg/μL [20]. The amplification specificity of LAMP was extremely high because the primers bind six independent sequences of the target DNA [15,19]. These properties attracted us to explore the use of LAMP for identification of sweet potato noodle adulteration.

The format of the reference section was inconsistent.  Authors should carefully check it.

Answer: The references were checked thoroughly in accordance with the journal requirement.

Submission Date

04 May 2019

Date of this review

19 May 2019 17:05:46